# Aerosol tracer testing in Boeing 767 and 777 aircraft to simulate exposure potential of infectious aerosol such as SARS-CoV-2

Sean M. Kinahan[1,2]ʘ*, David B. Silcott[3]ʘ, Blake E. Silcott[3‡], Ryan M. Silcott[3‡], Peter J. Silcott[3‡], Braden J. Silcott[3‡], Steven L. Distelhorst[2‡], Vicki L. Herrera[1‡], Danielle N. Rivera[2‡], Kevin K. Crown[2‡], Gabriel A. Lucero[2‡], Joshua L. Santarpia[1,2]

**1** University of Nebraska Medical Center, Omaha, Nebraska, United States of America, **2** National Strategic Research Center, Omaha, Nebraska, United States of America, **3** S3I LLC, Reisterstown, Maryland, United States of America

ʘ These authors contributed equally to this work.
‡ These authors also contributed equally to this work.
* skinahan@nsri.nebraskaresearch.gov

**Data Availability Statement:** Data for all real-time sensors, test times, conditions, and per second particle counts available at Figshare.com via the

## Abstract

The COVID-19 pandemic has reintroduced questions regarding the potential risk of SARS-CoV-2 exposure amongst passengers on an aircraft. Quantifying risk with computational fluid dynamics models or contact tracing methods alone is challenging, as experimental results for inflight biological aerosols is lacking. Using fluorescent aerosol tracers and real time optical sensors, coupled with DNA-tagged tracers for aerosol deposition, we executed ground and inflight testing on Boeing 767 and 777 airframes. Analysis here represents tracer particles released from a simulated infected passenger, in multiple rows and seats, to determine the exposure risk via penetration into breathing zones in that row and numerous rows ahead and behind the index case. We present here conclusions from 118 releases of fluorescent tracer particles, with 40+ Instantaneous Biological Analyzer and Collector sensors placed in passenger breathing zones for real-time measurement of simulated virus particle penetration. Results from both airframes showed a minimum reduction of 99.54% of 1 μm aerosols from the index source to the breathing zone of a typical passenger seated directly next to the source. An average 99.97 to 99.98% reduction was measured for the breathing zones tested in the 767 and 777, respectively. Contamination of surfaces from aerosol sources was minimal, and DNA-tagged 3 μm tracer aerosol collection techniques agreed with fluorescent methodologies.

## Introduction

The current COVID-19 outbreak caused by the SARS-CoV 2 coronavirus reintroduces questions regarding transmission risk during travel; as countries, companies, and individuals reduced travel to contain the outbreak and reduce exposure. In the US, the Transportation Security Administration (TSA) screened over 70% fewer travelers during most summer

following DOIs: USTRANSCOM 767 Inflight Master Spreadsheet with raw data. https://doi.org/10.6084/m9.figshare.13537319.v1 USTRANSCOM 777 Inflight Master Spreadsheet with raw data. https://doi.org/10.6084/m9.figshare.13537349.v1 USTRANSCOM 767 Hangar Master Spreadsheet with raw data. https://doi.org/10.6084/m9.figshare.13537358.v1 USTRANSCOM 777 Hangar Master Spreadsheet with raw data. https://doi.org/10.6084/m9.figshare.13537379.v2 USTRANSCOM 767 Terminal Master Spreadsheet with raw data. https://doi.org/10.6084/m9.figshare.13537367.v1 USTRANSCOM 777 Terminal Master Spreadsheet with raw data. https://doi.org/10.6084/m9.figshare.13537385.v1.

**Funding:** These studies were sponsored by United States Transportation Command (USTRANSCOM) through an Army Research Office Contract (W911NF-17-C-0060) to Zeteo Tech Incorporated. Zeteo Tech provided project management, programmatic support, and technical expertise. Additional test plan review and expertise were provided by DARPA Biological Technologies Office (BTO) as a courtesy to USTRANSCOM. United Airlines won a competitive bid for initial testing and methods development, and then donated fuel, airplanes, and flight time for the follow-on testing presented here at no cost. Boeing answered engineering questions and provided operational feedback on ECS system performance at no cost as a courtesy to USTRANSCOM. Instruments were loaned by DHS Science and Technology Directorate via MIT Lincoln Laboratories, and the National Guard Bureau via L2 Defense. NSRI, S3I, and UNMC received subcontracting funding from Zeteo Tech to perform experiments and provide analysis and aerobiology expertise. The funder provided support in the form of salaries but did not have any additional role in the study design, data collection and analysis, decision to publish, or preparation of the manuscript. The specific roles of these authors are articulated in the 'author contributions' section.

**Competing interests:** S3I and Zeteo Tech Incorporated both operate in aerosol instrumentation and detection development, and their affiliation as private companies does not alter our adherence to PLOS ONE policies on sharing data and materials. No author has a previous or current competing interest with Boeing, United Airlines, or the data and instrumentation utilized herein.

months of 2020 than a year earlier [1]. The CDC has determined that airborne transmission of SARS-CoV-2 can occur in environments which include enclosed spaces, prolonged exposure, and inadequate ventilation [2], which raises questions as to prolonged contact in an aircraft cabin environment. While projectile droplet transmission, dominated by their own inertia, likely does not vary substantially from the unique circumstance of being on an aircraft, the aerosol exposure route for smaller droplets and evaporated droplet nuclei is dependent on the ventilation, environment, and exposure time [3, 4]. A variety of research supports this conclusion, including sampling infectious virus in the air [4, 5], and case studies suggesting aerosol transmission [6–8]. An example of the latter occurred in a poorly ventilated nursing home unit where 81% of patients and 50% of staff became infected. The unit was equipped with an energy-efficient HVAC unit which only introduced fresh air when $CO_2$ levels climbed [8].

Quantifying the risk of inflight transmission is important for fully engaging economies, tourism, and business travel with human-to-human interactions. Since the exposure dose of any contaminant is a combination of the concentration and the length of time of exposure, it is important to understand both how quickly contaminants are removed, and what percentage of material penetrates the breathing zone of passengers. Existing strategies to evaluate pathogen risk on airlines include numerical models, such as computational fluid dynamics, and epidemiological studies of known-infected travelers and risk analysis.

Experimental data in these unique circumstances is challenging, in part due to challenges with methodology and instrumentation, and is largely completed in shorter cabin mockups. Tracer gases, such as sulfur hexafluoride ($SF_6$) or $CO_2$, have been used to determine airflow, with implications for airflow, gaseous contaminants, and small particulates that may more easily follow air streamlines [9–11]. Particle Image Velocimetry (PIV), stereoscopic PIV and volumetric particle tracking velocimetry (PVTV) have been used to examine both particle fate and visualization to help validate CFD models [9, 11–15].

Bennet, *et al.*, examined bacterial dispersion in an aircraft mockup, using a *Lactobacilli* spray bottle, and found a near-field and far-field cutoff that likely correlates to large and small-droplets, that follow a projectile pathway or become aerosols, respectively [16]. Li, et. al., utilized a combination of SF-6 and 3 μm low-volatility droplets for particle tracing in a retired MD-82 aircraft, and found agreement for 3 μm and below particulate with a gaseous tracer, with a rapid drop to nearby seats, and limited but varied longitudinal spread, that is forward or aft in the plane [17]. Longitudinal spread of contaminant was dependent on which tracer was utilized and where it was released. Overall, experimental data typically compares concentrations relative to other locations in the airframe or cabin mockup, with the source and quantity of contamination not characterized.

Epidemiological studies include survey and interview follow-ups for confirmed cases of both SARS-CoV-1 and SARS-CoV-2 in air transport. During the SARS-CoV-1 outbreak there was a series of three flights with infected passengers examined by Olsen, *et al.* [18]. One flight, 3 hours from Hong Kong to Beijing was indicative of transmission and resulted in 22 secondary cases from a single primary case. The other two flights, 90 minutes each, had a combined single possible transmission [18].

Studies from the ongoing COVID-19 pandemic indicate varied but generally less transmission during air travel. In one study, no secondary cases were traced on a 350-person 15-hour flight from Guangzhou, China to Toronto, Canada, which included a symptomatic (coughing), PCR-positive patient, and his wife, who tested positive a day after landing [19]. In another flight, 102 passengers traveled 4.66 hours from Tel Aviv, Israel to Frankfurt, Germany with 7 patients from a tourist group whom tested positive upon arrival. In this case, two in-air transmissions were possible, with both seated within 2 rows of an index case [20]. Whether transmission occurred via large-droplets, contaminated surfaces, or aerosol inhalation is informed

conjecture for these types of case studies. An ideal case study on an 18-hour Boeing 777 flight was completed in part thanks to the unique pre-testing, and quarantining required by New Zealand. During this flight, which included a stop for refueling (with the air system disabled) and in-flight meals, 4 in-flight transmission events occurred amongst 14 passengers located within 3 rows of an index case [21].

In this study, aerosol dispersion and deposition in two wide-body aircraft (Boeing a 767–300 and Boeing 777–200) was measured using fluorescent and DNA-tagged microspheres. Fluorescent 1 μm and DNA-tagged 3 μm particles were released and measured in multiple rows and seats distributed throughout each aircraft. Experimental data included over 300 releases from a simulated SARS-CoV2-infected passenger in seats on 767–300 and 777–200 airframes. This manuscript focuses on the 118 in-flight releases of a simulated breathing condition, where extended exposure times are most likely. The tests were designed to measure the aerosol concentration within passenger breathing zones in neighboring seats and rows from the simulated infected passenger. The tests were also designed to measure passenger breathing zone aerosol concentration distributions at different sections of the airframes and with the simulated infected passenger seated at various locations, to determine if differences in exposure risk existed for different seats or rows of the aircraft. Additional testing, performed on the ground or at the terminal is the focus of follow-on studies [22].

## Materials and methods

### Airframe testing

Testing of each airframe totaled four days, with two days reserved for ground testing (not discussed here), and two days reserved for in-flight testing at altitude (typically 30,000+ feet). Only in-flight data is presented here, but data for all testing is being made available on Figshare, including limited seats and releases tested at an accelerated release velocity, not analyzed here, to more closely mimic coughing (S7 & S8 Tables). Testing occurred at Dulles International Airport (IAD) with the first four days reserved for the Boeing 777, and the second four reserved for the Boeing 767.

Air exchange rates for the specific 767 and 777 airframes tested were reported to be 32 and 35 air changes per hour (ACH), respectively, with total cabin volumes of 263.9 and 426.9 cubic meters. Both Environmental Control Systems (ECS), responsible for pressurization and cabin air supply, achieve approximately 50% of the air exchange through HEPA-filtered recirculation, and 50% through fresh air pressurized from the engine. Airflow is designed to flow from top to bottom, with minimal mixing forward or aft, so that any contaminants would ideally not spread to nearby rows. Air is supplied above the head, and exhausts at the feet on the sides of the airframe, with flow rate matched locally to limit movement up and down the plane (Fig 1). The air velocity is minimal, with typical aircraft designs requiring a range from 0.005 and 0.3 m/s, to limit the feeling of stagnant air or a breeze [15, 23]. Although personal gasper position was studied during ground testing, many aircraft do not have gaspers available, and in-flight testing conditions were limited. Other studies and existing modeling suggest their effect on infection risk is neutral [24]. Consequentially, they remained in the off position for in-flight testing.

Instantaneous Biological Analyzer and Collectors (IBAC, FLIR Systems) sensor layouts and release locations for each airframe and section tested are available in S1 Fig. The sections were intended to distribute releases evenly throughout the airframe, with multiple sections in economy seating. These sections were named for their relative position in the airframe (Forward, Forward-Mid, Mid-Aft, and Aft on the 777 and Forward, Forward-Mid, and Aft on the shorter 767). Although a single release seat is marked, in all real-time fluorescent test cases multiple releases were completed at multiple seats in a row throughout each section. An overview of

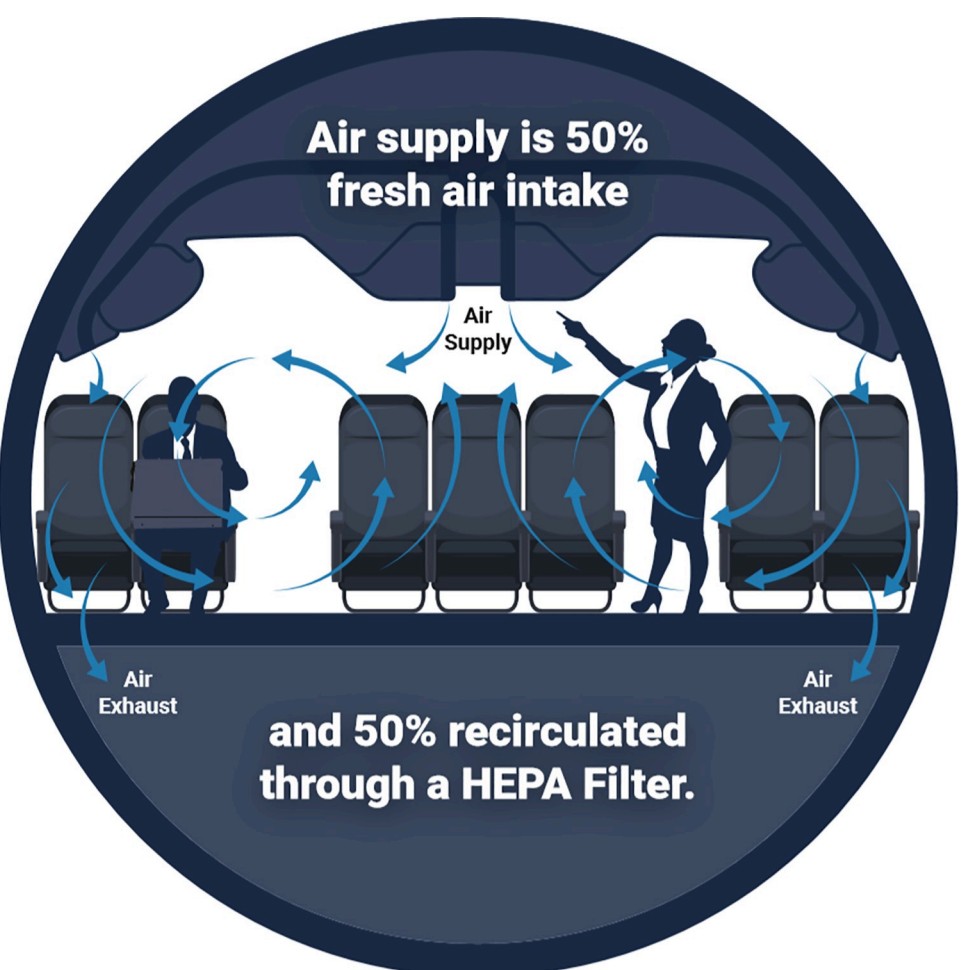

**Fig 1. Notional diagram of airflow in an aircraft cabin (767).** A mix of HEPA-filtered recirculated air and fresh pressurized engine bleed air enters at the top, mixes and exhausts through the bottom. The system is designed to minimize forward or aft mixing along the length of the plane.

test days is described here, with specific dates, conditions, seat positions, and variables is in the supporting information (S1–S8 Tables).

### 777–200 testing

During the two days of in-flight testing, fluorescent tracer particles were released in the AFT, MID-AFT, FWD, and FWD-MID sections of the airframe for a total of 64 releases. The breathing releases included 39 fluorescent tracer tests, described below, with the mannequin not wearing a mask and 24 tests with a mask. Limited by the amount of test time available, multiple seats were prioritized over testing the mask at every seat, and up to three seats were tested per row. Seats included 47B, 47E 47K, 33B, 33E, 33K, 11A, 11G, 11L, 5A, 7A, 5G and 5L. Triplicate releases were performed for each mask on/off condition.

### 767–300 testing

Fluorescent tracer particles released in the AFT, FWD-MID and FWD sections of the airframe for a total of 55 releases from a breathing simulation during two days of inflight testing, 27

without a mask, and 28 with a mask. Specific seats included 37B, 37E, 37K, 18A, 18E, 18L, 6A, 6D, and 6L. Releases were completed in triplicate, with a detailed testing breakdown in the supporting information (S1–S8 Tables). Throughout testing, the only technical issues were occasional loss of power to IBAC sensors due to loose connection to an airframe outlet, resulting in some locations missing data for a subset of releases.

## Chamber characterization and source terms

To better understand the tracer releases, four IBAC instruments and three TSI Inc. 3321 Aerodynamic Particle Sizers (APS) characterized the tracer releases with and without a mask in a characterized, controllable aerosol chamber. The chamber is a High Efficiency Particulate Air (HEPA)-filtered, rapidly-purged test chamber, where naturally-occurring background aerosols are minimized. During a test, the chamber was purged of particulate for two minutes, and then placed into a static mode without airflow. The 1 μm test solution, $4.55^*10^9$ particles/ml in deionized water, was nebulized identically to in-flight tests (described below), from a nebulizer plumbed to a mannequin (fluorescent tracer) with or without a mask. The DNA-tagged tracer was aerosolized solely without the use of a mask, and in both cases the chamber and particulate are briefly mixed using two remote-controllable fans (20–25 seconds).

The aerosol instrumentation characterized the resulting aerosol in particles per liter of air (pla) (S2 Fig). At 11902 liters, the average concentration across the aerosol detectors is multiplied by that total volume to give the amount of tracer particulate released and verify the size distribution (S3 Fig). The total number of particles for each release condition has low standard error around the mean, with a total number of fluorescent tracers when unmasked of $1.8^*10^8$ (with mask $1.7^*10^8$), compared to $2.4^*10^7$ beads for the larger 3 μm DNA-tagged tracers (Table 1). The total number of particles released is essential for comparison with how much particulate enters the breathing zone of a given location.

The test sizes were chosen based on existing understanding of sizes most likely to contain SARS-CoV-2 virus. Liu, *et al.* sampled for RNA in Hospitals and found bimodal results in submicron (0.25 to 1 μm) and supermicron (> 2.5 μm) ranges [25]. For comparison, a multimodal aerosol generation for typical persons breathing nose to mouth could be fit with four modes, with diameters of 0.8 (86%), 1.8 (9%), 3. (3%) and 5.5 (2%) μm [26]. Importantly, the physical size of SARS-CoV-2 has been reported as 50–200 nm [27], and while single virus particles without accompanying phlegm are unlikely, additional research defining the infectious particulate generated by patients is needed. As particle sizes become increasingly sub-micron

**Table 1. Controlled chamber aerosol characterization.** Each type of tracer release was characterized in triplicate, including the total 1 μm fluorescent particles with and without a mask, and 3 μm DNA-tagged particles (without a mask only).

| Chamber Characterization | Breathing without Mask | Sample Size (n) | 3 |
| --- | --- | --- | --- |
| | | Mean (Total Particles) | 1.80E+08 |
| | | Standard Deviation | 1.70E+07 |
| | | Standard Error of Mean | 1.00E+07 |
| | Breathing with Mask | Sample Size (n) | 3 |
| | | Mean (Total Particles) | 1.70E+08 |
| | | Standard Deviation | 8.50E+06 |
| | | Standard Error of Mean | 4.90E+06 |
| | DNA-Tagged (without mask) | Sample Size (n) | 3 |
| | | Mean (Total Particles) | 2.40E+07 |
| | | Standard Deviation | 4.30E+06 |
| | | Standard Error of Mean | 2.50E+06 |

the effects of inertial forces vs Brownian motion will shift, and physical transport will vary more from our larger test aerosols [28].

## Fluorescent tracer aerosol detection

A suite of IBAC sensors, discrete particle detectors that simultaneously measures an airborne particle's elastic scatter and fluorescence at an excitation wavelength of 405 nm, provided tracer detection and quantification. The IBAC contains two fluorescence thresholds, one for biological aerosols and the other for fluorescent tracer aerosol detection. IBAC sensors have been used for fluorescent tracer particle dispersion exposure testing and mapping in government, research, and clinical settings including subway systems, airports, skyscrapers, large building complexes, critical infrastructure facilities, commercial aircraft and numerous other types of buildings [29–32]. Although historically predominantly used by clinical environments and government applications, tracer studies offer a methodology to characterize and quantify exposure risk in other typical infectious environments which are needed during the pandemic, such as restaurants, gyms, and homes.

Fluoresbrite 1 μm yellow-green (YG) polystyrene latex (PSL) microspheres (Polysciences), with a fluorescent signal distinct from naturally-occurring aerosols served as the test tracer. The comparatively fluorescent background particulate in a natural test environment, including airframes is negligible. During these tests, the background concentration was typically <5 pla, or 100 particles over a 6-minute integrated test.

The instrument samples at 3.5 liters per minute (lpm), and reports tracer counts per second. Prior to the airframe tests, the 42 IBACs were calibrated and the fluorescent particle tracer counts were matched to within an average variance of ±10%. IBAC sensors were placed in individual seats, with either 1 foot tubing extensions to actively sample within the breathing zone or by placing on a collapsible crate to achieve the same sampling location (S4 Fig), using omni-directional inlets. These were left to sample at 1-second intervals until rearranged for the next seating, with a portion of the systems connected for real-time monitoring of return to baseline between tests, which occurred within 8 minutes, depending on location. The inlet therefore samples from where a passenger's head and mouth would be located when seated in flight. The low air velocities dictated by passenger comfort limited the need for isokinetic sampling or inlet alignment with a flow field.

Particulate penetration percentage was calculated as the number of particles observed in each seat integrated over the release duration and return to background, normalized to a typical resting passenger respiratory minute volume (7.5 lpm) rather than the instruments sampling rate of 3.5 lpm, and then divided by the total number of particles released based on the chamber characterization ($1.8^*10^8$). This ratio represents the percentage of particulate, or mass for this mono-dispersed release, a person seated at rest near a source would likely inhale.

## DNA-tagged microspheres

DNA-tagged tracers were streptavidin-coated 3 μm PSL microspheres (Bangs Laboratories) bound with four unique 5'-biotynlated DNA oligos. Each oligo was designed to be 170 base pairs in length, non-coding and confirmed not to match natural sequences using a Basic Local Alignment Search Tool (BLAST) search [33]. Complimentary quantitative real-time polymerase chain reaction (qRT-PCR) assays were designed for detection (IDT Inc.) targeting a 60˚C extension and anneal step. Binding of biotinylated DNA occurred per the manufacturer's protocol, scaled to a 3 ml production volume, with the test particles washed five times via centrifugation at 10,000 rpm to ensure removal of any unbound DNA.

Standard curves were developed for each bound oligo to quantify the number of beads, using a 40 cycle 95˚C melt, and 60˚C anneal and extension protocol on a QuantStudio 3 (ThermoFisher Inc). All samples were processed in triplicate, with dilutions of positive and negative controls in parallel, and each oligo used a uniform threshold for detection. No cycle threshold's (Ct) above background negative controls were accepted, and at least two of three replicates were required to be positive for analysis. This approach mimics PCR-based encapsulation of DNA-targets utilized by Harding, *et al.* for tracer experiments [34].

## Aerosol and surface collection

DNA-tagged tracers were collected at 50 liters per minute using an Airport MD8 aerosol sampler (Sartorius), which operated for fifteen minutes, and has been shown to collect 96+% of *Bacillus* spore particles between 0.7 and 1.0 μm [35, 36]. Gel filters were extracted into 15 ml of deionized water, vortexed for 30 seconds, and diluted 1:10 in nanopure water for qPCR analysis. Five of these air collectors were distributed near release rows and in the galley, configured to sample facing into the breathing zone of each seat.

Surface coupons consisted of 8.89 cm long, 2.54 cm wide (0.6 mm thick) stainless-steel rectangles held using painters tape, leaving a total area of 16.13 cm$^2$ exposed during a release. These coupons were aseptically collected into 50 ml conical tubes, suspended using deionized water (10 ml), vortexed for 30 seconds, then utilized for qPCR. In between tests, areas were wiped using DNAaway and deionized water to remove any carryover. Coupon locations targeted common touch surfaces including arm rests, tables, and seatbacks (S5 Fig).

DNA-tagged beads were released in flight from three 767 locations (forward, mid-forward, and aft) and three 777 locations (forward, mid-forward, mid-aft), with surface coupons dispersed near the release seats, to look at fomite risk from a passenger due to aerosol surface deposition. Testing was completed in triplicate and averaged. Cycle thresholds converted into a number of beads per ml based on the qPCR standard curves. The concentration is transformed to a total number of beads based on the volume of the sample and the dilution. Comparing the number of beads collected at an aerosol collector to the total number released based on chamber characterization, gave a percentage of the total number of tracers that settled. In the case of surface samples, where the number of beads is per unit area, the percentage of beads captured at each location is scaled to a one square foot standard area.

## Nebulization & face mask

Either a Devilbiss Traveler (DNA-tagged tracer) or Devlibiss PulmoMate (fluorescent tracer) nebulizer provided aerosolization. DNA-tagged beads were aerosolized for five minutes and fluorescent tagged microspheres were aerosolized for one minute in a breathing pattern for 2 seconds on and 2 seconds off. The output of the nebulizer cup (Hudson Micro Mist) is plumbed through the mouth of a tripod mounted mannequin head (S4 Fig) and reaches a velocity of 1.43 m/s at the mannequin's lips. The mannequin was used to allow for control of velocity of output air, to locate the release in the breathing zone of a passenger, and to incorporate testing of a facemask.

Standard disposable 3-ply masks (Guandong Paso Automible Technology Co., Ltd), which are the most likely to be handed out by an airline, were tested. One pandemic survey suggests that in the US, cloth masks were most worn, at least weekly by participants at 75%, but 3-ply masks were next most common at 57% of participants engaging in weekly use and this style is given out by airlines [37].

**Table 2. Particle penetration into nearby seats on a Boeing 767 and Boeing 777 aircraft.** Economy (particles generated through the mannequin with and without a mask) and first class penetration percentages for nearby seats on the Boeing 767 and 777. Nearby seats are defined as within 2 seats left, right, front and back, and the four closest diagonal seats.

| | | | | Economy | | First Class |
|---|---|---|---|---|---|---|
| | | | | **Breathing w/o Mask** | **Breathing w/ Mask** | **Breathing w/o Mask** |
| Airframe | 767 | Penetration Percentage (%) | Sample Size (n) | 177 | 172 | 54 |
| | | | 95.0% Lower CL for Mean | 0.0148% | 0.0133% | 0.0099% |
| | | | Mean | 0.0176% | 0.0158% | 0.0119% |
| | | | 95.0% Upper CL for Mean | 0.0205% | 0.0183% | 0.0139% |
| | | | Standard Deviation | 0.0190% | 0.0165% | 0.0074% |
| | | | Standard Error of Mean | 0.0014% | 0.0013% | 0.0010% |
| | | | Minimum | 0.0000% | 0.0000% | 0.0023% |
| | | | Median | 0.0127% | 0.0107% | 0.0110% |
| | | | Maximum | 0.0947% | 0.0796% | 0.0337% |
| | 777 | Penetration Percentage (%) | Sample Size (n) | 152 | 120 | 102 |
| | | | 95.0% Lower CL for Mean | 0.0176% | 0.0120% | 0.0120% |
| | | | Mean | 0.0262% | 0.0160% | 0.0150% |
| | | | 95.0% Upper CL for Mean | 0.0348% | 0.0201% | 0.0180% |
| | | | Standard Deviation | 0.0538% | 0.0223% | 0.0153% |
| | | | Standard Error of Mean | 0.0044% | 0.0020% | 0.0015% |
| | | | Minimum | 0.0000% | 0.0000% | 0.0000% |
| | | | Median | 0.0154% | 0.0081% | 0.0094% |
| | | | Maximum | 0.4614% | 0.1157% | 0.0614% |

## Results

### Fluorescent tracer results—777

Using the particle penetration percentage based on a 7.5 lpm minute volume, the maximum exposure, 0.4614% occurs in a seat next to a source, and occurred during one release from seat 33B on the Boeing 777 (Table 2). The next two highest penetration percentages also occurred at this release location, and corresponded with 0.2577% and 0.3737%. These were extreme outliers compared to all other releases on the 777 airframe. For a typical seat located immediately nearby a source, that is in front, behind, diagonally in front or behind, and two seats in front, behind, or left and right, the 95% confidence interval (CI) for mean particle penetration percentage was 0.0176% to 0.0348%. Averaging across all sensors in the AFT and MID-AFT economy sections, the mean penetration percentage was 0.0124% and 0.0118%.

Examining Fig 2, although overall penetration percentages are low for nearby seats, and there are trends for seats adjacent, front, or behind a simulated infected passenger, it does depend on the release seat on the airframe (Fig 2A). The seats next to a release were naturally the highest, with the next highest typically occurring in the seats behind the simulated infected passenger. When grouping these releases in economy section of the 777, there was a statistically significant difference between groups as determined by one-way ANOVA ($F_{(7,144)}$ = 4.139, p = 0.000). Following that with a Tukey HSD post-hoc analysis and comparing with Fig 2A, indicates that the largest statistical differences are between the seats immediately next to the release point, which are statistically significant when compared to the diagonal front seats, front seats, and seats two away (p-values of 0.000, 0.023, and 0.029, respectively).

Notably, the differences between a seat in front, and diagonally in front, or rearward and diagonally rearward, are small, indicating that mixing within a row is occurring rapidly (Fig 2A and 2B) Plotting the average penetration percentage across all sensors in a row, quantified

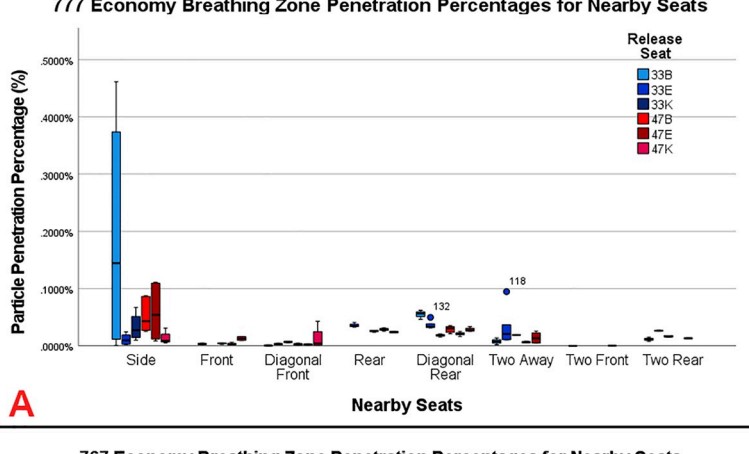

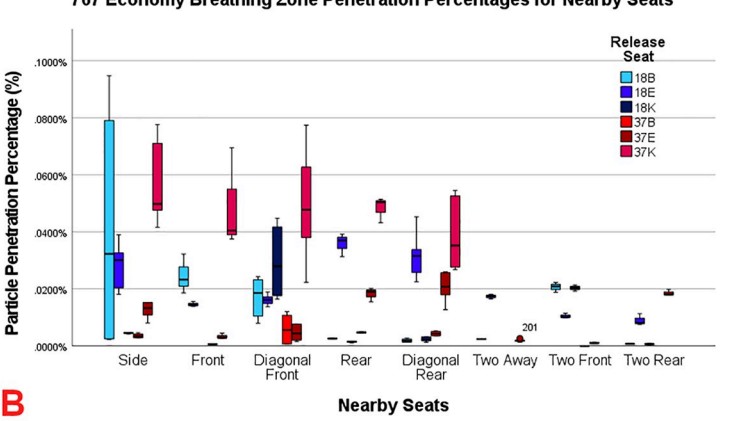

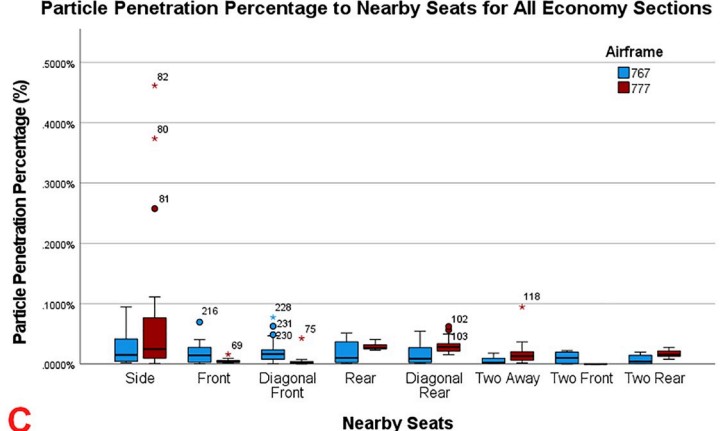

**Fig 2. Particle penetration percentages for various 777 and 767 seat configurations.** A) Penetration percentages for breathing (2 s on and 2 s off) releases with the unmasked mannequin in seats nearby a thesource throughout an economy section of a Boeing 777. B) Penetration percentages for breathing unmasked mannequin releases throughout an economy section of a Boeing 767. C) Penetration percentages for releases throughout an economy section of a Boeing 777 compared to a 767. The 777 side seats have three outliers not presented in C) representing the maximum exposure for all experiments (0.2577, 0.3737, and 0.4614%).

the relative longitudinal flow forward or aftward depending on the release location. Fig 3A shows that for both economy sections and release seats tested during 777 flights, contaminants mix towards the aft of the plane, where the outflow valve is located.

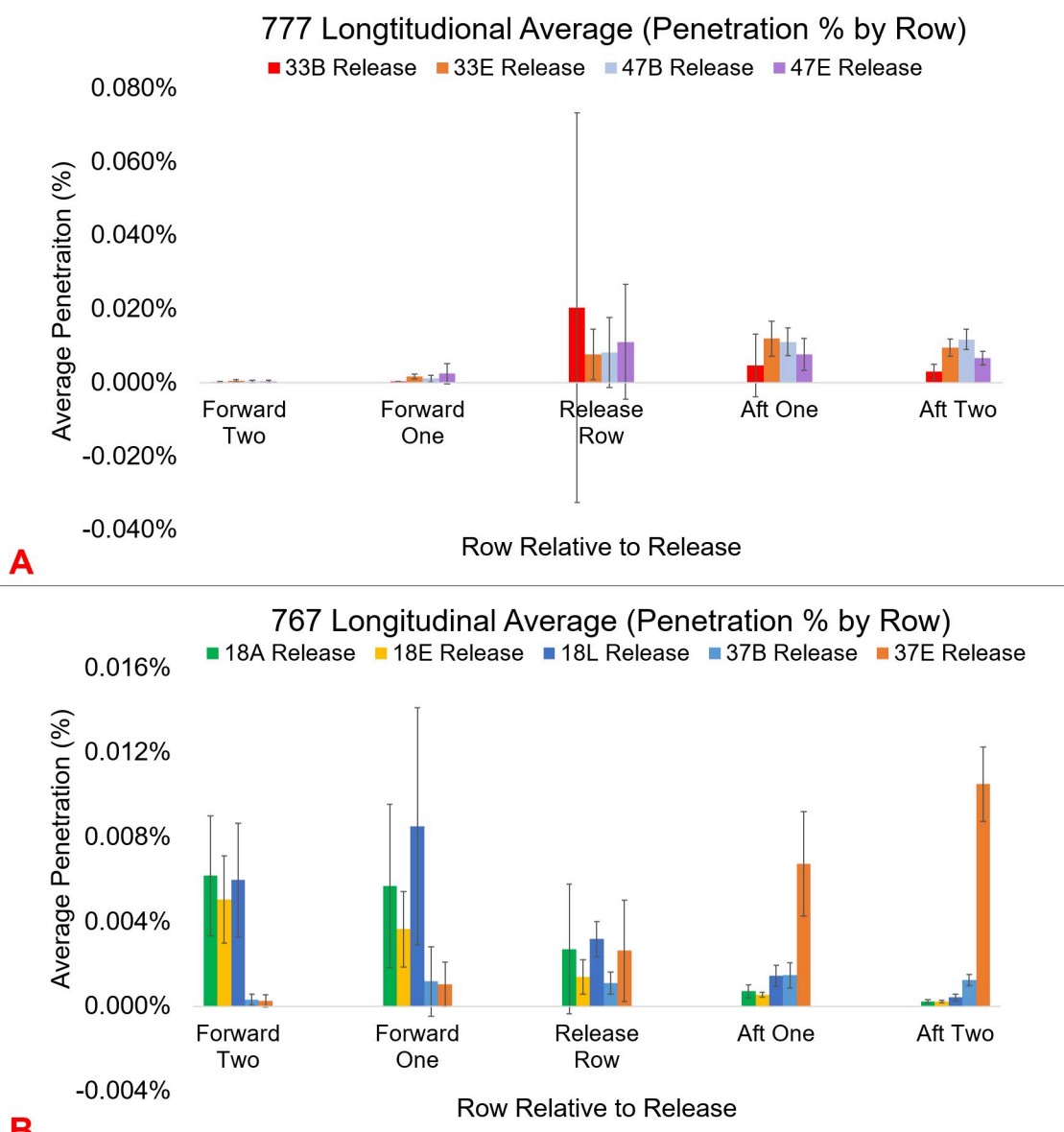

**Fig 3. Comparison of longitudinal movement of particles forward and aft of the release.** A) 777–200 longitudinal average penetration percentage (error bars represent one standard deviation) across all sensors in a given row. For the 777 there is a clear aft movement of contaminant from both row 33 and row 47. The release row contains the highest uncertainty, as seats next to the release point are averaged with sensors much further in the same row. B) 767–300 longitudinal average penetration percentages across sensors in a given row. For the 767 forward releases from row 18 mix forward, for aft releases, contamination mixes towards the outflow valve in the rear of the airframe.

First class seats do not have the same arrangement, with nearby passengers inherently more spread out. The maximum exposure percentage (0.0614%) in the forward and fwd-mid (rows 5 and 11, respectively) occurred in a seat (12A) immediately behind the release location (11A). The 95% confidence interval for the mean exposure risk for those same closest seats on this airframe ranges from 0.0120 to 0.0150%. Amongst all the sensors spread out throughout the section, the average exposure risk for the fwd-mid by penetration percentage was 0.0026% and for the forward section, 0.0034% (Table 2).

## Fluorescent tracer results—767

The maximum exposure risk of the Boeing 767 was lower than the 777, measured at a penetration percentage of 0.0947%. The highest risk was measured in a seat (18A) located immediately adjacent to a release location for an infected passenger (18B), with the next highest penetration percentage also measured in seat 18A, at 0.0791%. However, when averaging those seats closest to an infected passenger, two seats away (front, back and to the side) and one seat diagonally, the 95% CI for the mean penetration percentage was from 0.0148 to 0.0205% (Table 2). For the 767 in-flight tests, the seats closest to the release remained the most at risk for inhalation, but there were no extreme outliers, and the importance of the specific release seat on the airframe becomes apparent (Fig 2B). Again, the rapid mixing, and high air exchange rate led to seats diagonally located away from the release to behave similarly as those immediately in front of the release, and those located behind the release to behave similarly to those located diagonally behind (Fig 2B and 2C). When averaging penetration efficiency amongst every seat with a sensor in the section, the mean penetration efficiency decreases to 0.0088 and 0.0086% for the aft and fwd-mid sections of the airframe.

In the 767, combining release location data to examine nearby seat relationships leads to expanded box and whisker plots (Fig 2C). The corresponding ANOVA concluded that there is a statistically significant difference between groups ($F(7,169) = 2.718$, $p = 0.011$, alpha = 0.05), but the post-hoc Tukey HSD, determined that the statistical significance is limited to seats located to the side of a release, and those located two seats away. Judging from the individual seat release comparisons, there is further variance based on the location within the plane (Fig 2B) when compared to the 777 (Fig 2A).

Comparing the longitudinal flow for the 767, there is less aftward flow of contamination. The aft release seats in row 37 primarily mix towards the aft of the airframe, but the fwd-mid releases in row 18 primarily mix forward (Fig 3B).

First Class Seats in the Boeing 767 experienced lower maximum and average penetration percentages, with the two maximums, 0.0337 and 0.0305%, occurring in seat 7A, diagonally behind the release seat 6D. The 95% CI for the mean exposure risk in the nearby seats ranged from 0.0099 to 0.0139%, but importantly, there were no IBAC sensors directly next to any of the release locations in this forward first class section. Amongst all nearby sensors in the forward section, the average was 0.0047% (Table 2).

## Effect of mannequin mask

The efficacy of the standard 3-ply mask was not the primary goal of this testing as the variety of droplet sizes and velocities at the mouth vary at the mouth/mask interface, especially as a function of whether they are generated via talking, breathing, coughing, or sneezing. This group of tests was rather to determine if redirecting air at lower breathing velocities with a facemask in place made a difference in the number of particles reaching the breathing zone of nearby passengers. Existing literature visualizing droplet and jet production with a mask found leakage and redirection of jets and particulate upwards, downwards, and to the sides of the mask [38, 39].

For each set of tests with and without a mask in the same seat, total particle counts at all sensors were compared, to see whether the exhaled particles were redirected elsewhere in the plane. There is a statistically significant difference between tests from the same seat with and without a mask ($p = 0.045$, alpha 0.05), when using a paired student's t-test. The average reduction with a mask in total particles counted at all sensors was 15.6%, compared with a mean reduction of 7.6% in the static characterization chamber (Table 3). However, given the dynamic way these particles are mixing in the cabin, there was a large standard deviation.

**Table 3. Comparison of triplicate particle counts for all sensors during 14 economy seat releases with and without a mask.** A total of 14 release seats in economy were compared. On average the mannequin mask reduced the total particles in nearby breathing zones by 15.6%, compared to not wearing a mask.

| | | Statistic | Result |
|---|---|---|---|
| In flight Mask Reduction | Reduction in Total Particles for a Release Seat (%) | Sample Size (n) | 14 |
| | | 95.0% Lower Confidence Interval for Mean | 3.50% |
| | | Mean | 15.60% |
| | | 95.0% Upper Confidence Interval for Mean | 27.70% |
| | | Standard Deviation | 20.90% |
| | | Standard Error of Mean | 5.60% |
| | | Minimum | -26.6% |
| | | Median | 17.70% |
| | | Maximum | 52.30% |

Further, there was an increase in total particle counts at times, with a maximum increase of 26.6% and a total of 3 of the 14 release seats showing an increase (767 seats 18L and 18E, 777 seat 33E) in total counted particles in all breathing zones.

As further evidence of the mannequin's mask redirecting particles rather than filtering them, of the 11 source seats with total particle count decreases, 9 sensors (nearby seats) on average observed an increase in particle count, with 27 on average observing a decrease. Importantly, the fit of the mask to a mannequin face may vary from a human being, and increase noise if fit varies from location to location or test to test.

## DNA-tagged results– 777

DNA-tagged microspheres demonstrate clear trends similar to real-time data. In the case of air samples, the collected fraction of particles aerosolized compares well with the real-time fluorescent tracer, ranging from undetectable to 0.030% in economy sections closest to the release point (Fig 4A). The highest collected aerosol concentration was always located closest to the release point of that DNA-tagged bead, with lower risks forward of a release than aft of the release. A low concentration of tracer particles were present in the aft galley from the economy release, and the aftward movement of contaminant seen in the fluorescent particulate data is duplicated here.

Surface samples located on the arm rests and seat backs of the seats closest to each release location were scaled from their size to a standard square foot for better comparison. Even scaling to a larger surface area, less than 0.03% of tracer particles settle out during testing, with the highest concentration on the surfaces closest to each release location. Notably horizontal surfaces, such as arm rests were typically higher than vertical surfaces such as seatbacks and inflight entertainment (IFE) systems. The low overall deposition leads to higher 95% CI (S9 Table).

## DNA-tagged results– 767

The DNA-tagged tracer releases completed on the 777 were duplicated on the 767, with surface samples targeted at high-touch surfaces. Like the 777, air samplers agreed with the fluorescent real-time releases, with the highest number of particles nearest each release location, and the overall percentage of particles compared to the chamber characterization consistently below 0.02% located 3 rows away (Fig 4B).

The percentage of particles that settled onto contaminated surfaces, scaled to a standard square foot, remains low with a maximum deposition percentage less than 0.005%. Arm rests and table tops closest to the release location consistently had the highest level of contamination

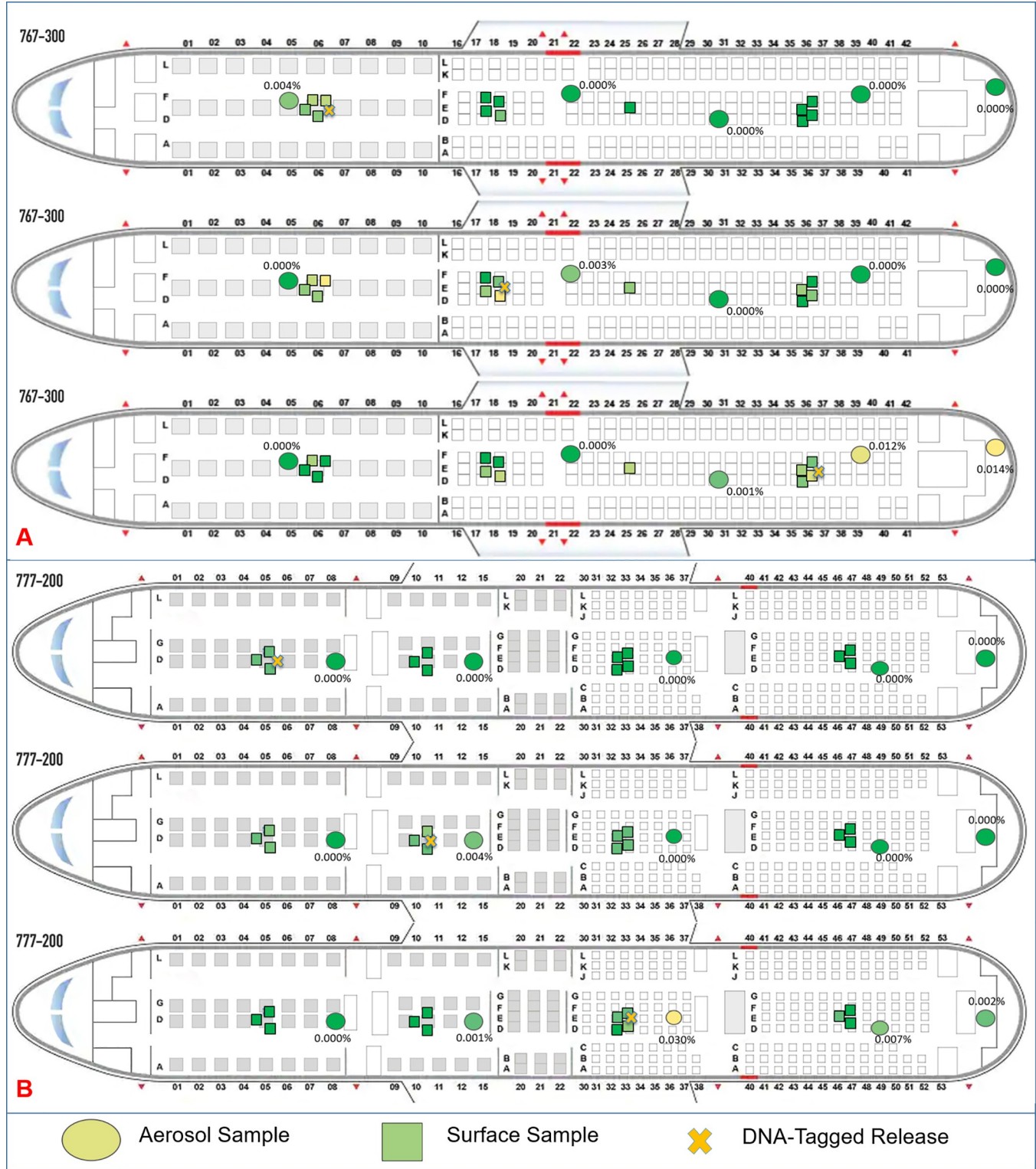

**Fig 4. DNA-tagged tracer maps.** A) 777–200 DNA-tagged tracer particle maps for fwd, fwd-mid, and mid-aft releases. B) 767–300 DNA-tagged tracer particle maps for fwd, fwd-mid, and aft releases. Surface contamination is minimal, and aerosol collections are similar to real-time fluorescent results. Circles represent aerosol samples, with squares representing surface samples. Colors are assigned solely for relative comparison purposes.

for each release location. Confidence intervals are large for surface samples due to low overall deposition and signal (S10 Table).

## Discussion

Overall, rapid mixing, dilution and removal limit exposure risk for aerosol contaminants at 1 and 3 μm in all tested seat sections of the Boeing 767 and Boeing 777 wide body aircraft. The maximum exposure in a nearby seat of 0.4614% of a characterized release, equates to a 99.54% reduction from an aerosolized source of contamination such as SARS-CoV-2. Looking further across the approximately 40 seats nearby the simulated infected patient there is average reduction maximum in the aft section of the 777, with exposure risk of 0.0124%, representing a 99.99% reduction. Importantly, this represents a single infectious point source, not a scenario with multiple infected passengers. Testing focused on aerosol transport and smaller 1 to 3 μm particulate. Larger droplets (10s to 100s of microns) generated and co-released with smaller modes when talking, coughing, or sneezing introduce an alternative transmission mechanism, which face masks have been shown to statistically reduce in other literature [40, 41].

Importantly, by using tracer to study transport, we are intentionally studying the physics of exposure. Chemical and biological processes also affect the infectivity and activity of any biological aerosol, especially a virus. SARS-CoV-2 has been shown to deactivate in the aerosol phase by a variety of processes, including UV irradiation and relative humidity. We focus on exposure risk, because infection risk must also consider environmental fate, infectious dose, and shedding rates [42, 43].

The infectious dose and viral load emitted by an infected passenger may also depend on their disease severity, length of infection, age and profile, many variables which are currently unknown. As an example, the number of particles emitted may vary substantially, with superemitters generating an order of magnitude or more aerosols than a typical person while talking or breathing, perhaps as many as 6000 in ten minutes of conversation [44, 45]. While the percentage of these that contain infectious material, and the infectious dose are still unknown, the purpose of this analysis is to demonstrate the risk to most passengers is low and decreases rapidly with distance. However, being seated next to shedding passenger likely still carries risk, especially depending on assumptions regarding these unknown variables. If a single particle contains an infectious dose, low penetration percentages would still be sufficient for those closest passengers to a source.

These data support the findings of epidemiologic studies of commercial airframe transmission for passengers with SARS-CoV-2, especially those that were unlikely to be exposed in other circumstances. Freedman & Wilder-Smith reviewed all known studies of passenger air travel with COVID-positive passengers that studied potential secondary cases [46]. Conclusions included a summary of 8 mass transmission flights, typically with many primary cases, and 58 cases with zero transmission. In another epidemiological study, 102 passengers traveled 4.66 hours from Tel Aviv, Israel to Frankfurt, Germany with 7 patients from a tourist group who tested positive upon arrival. In this case, two in-air transmissions were probably, with both seated within 2 rows of an index case [19].

This methodology similarly does not focus on contamination of surfaces via non-aerosol routes (via large droplets or fecal contamination), which is likely more common in areas such as the lavatory. Alternative routes of exposure are more challenging to predict because of uncertainty in human behavior [47]. Testing did not include movement of passengers throughout the plane, and the mannequin remained facing forward. Testing was also limited to two particles sizes, and three to four rows of each aircraft. Conclusions outside of specifically tested seats and breathing zones assumes extrapolation of the data to new conditions.

Widespread aerosol exposure risk is likely minimal during long duration flights, but still present, and is notably highest in the row of an index patient. Rows in front and behind the index patient have the next highest risk on average. While there is a measurable difference in middle vs aisle or window seat, results show that this relative exposure risk is not generalizable and depends on location and seat throughout the cabin.

## Supporting information

**S1 Fig. Sensor layouts.** IBAC sensor layouts for each airframe and section tested. A) 767–300 sections and seats B) 777–200 sensors and seats. A single release seat is shown, but releases were done in multiple seats within a given row.
(TIF)

**S2 Fig. Chamber characterization layout and sensors.** Chamber testing using a mannequin, three APS particle sizers, and four IBACs.
(TIF)

**S3 Fig. Characterization of aerosol tracer particles at 1 and 3 μm.** The aerodynamic size distribution for each tracer size normalized for width of size bin (dN/dlogDp).
(TIF)

**S4 Fig. Visualization of mannequin and IBAC placement.** Visualization of mannequin and instruments on a Boeing 767, with and without a mask.
(TIF)

**S5 Fig. Example sample coupon placement.** Coupons and locations highlighted in red. Left: Economy seat. Right: First class seat.
(TIF)

**S6 Fig.**
(DOCX)

**S1 Table. Boeing 777–200 test conditions and timeline for hangar.** Hangar testing for the Boeing 777–200 on August 24, 2020.
(DOCX)

**S2 Table. Boeing 777–200 test conditions and timeline for jetway.** Jetway testing for the Boeing 777–200 on August 25, 2020.
(DOCX)

**S3 Table. Boeing 777–200 test conditions and timeline for first day of inflight testing.** Inflight testing day 1 for the Boeing 777–200 on August 26, 2020.
(DOCX)

**S4 Table. Boeing 777–200 test conditions and timeline for second day of inflight testing.** Inflight testing day 2 for the Boeing 777–200 on August 27, 2020.
(DOCX)

**S5 Table. Boeing 767–300 test conditions and timeline for hangar.** Hangar testing for the Boeing 767–300 on August 28, 2020.
(DOCX)

**S6 Table. Boeing 767–300 test conditions and timeline for jetway.** Jetway testing for the Boeing 767–300 on August 29, 2020.
(DOCX)

**S7 Table. Boeing 767–300 test conditions and timeline for first day of inflight testing.**
Inflight testing day 1 for the Boeing 767–300 on August 30, 2020.
(DOCX)

**S8 Table. Boeing 767–300 test conditions and timeline for second day of inflight testing.**
Inflight testing day 2 for the Boeing 767–300 on August 31, 2020.
(DOCX)

**S9 Table. 777–200 DNA-tagged tracer results.** Large confidence intervals (n = 3, 95% CI based on standard error of the mean) reflect the low nucleic-acid signal.
(DOCX)

**S10 Table. 767–300 DNA-tagged tracer results.** Large confidence intervals (n = 3, 95% CI based on standard error of the mean) reflect the low nucleic-acid signal.
(DOCX)

## Acknowledgments

The team would like to acknowledge United Airlines for their donation of time, access, and expertise, including physical use of two airframes, access to a terminal jetway, crew, support for inflight testing, and engineering and technician support for electrical and ECS system performance. Thank you also for Boeing engineers who were available to answer significant questions about expected ECS system behavior and design, including providing air exchange and volume measurements. A special thanks also belongs to Zeteo Tech Inc, especially Wayne Bryden and Mike McLoughlin, for their technical and programmatic assistance with aerosol testing. L2 Defense and Russel Accardi helped facilitate loaning of detection instrumentation for experimentation. All statistical analysis and plots were completed using IBM Corp.'s SPSS for Windows, Version 27, Released 2020.

## Author Contributions

**Conceptualization:** David B. Silcott, Kevin K. Crown, Joshua L. Santarpia.

**Data curation:** Sean M. Kinahan, David B. Silcott, Blake E. Silcott, Ryan M. Silcott, Peter J. Silcott, Braden J. Silcott, Vicki L. Herrera, Danielle N. Rivera, Gabriel A. Lucero.

**Formal analysis:** Sean M. Kinahan, David B. Silcott, Blake E. Silcott, Ryan M. Silcott, Peter J. Silcott, Braden J. Silcott, Vicki L. Herrera, Joshua L. Santarpia.

**Investigation:** Steven L. Distelhorst, Vicki L. Herrera, Danielle N. Rivera, Kevin K. Crown, Gabriel A. Lucero.

**Methodology:** Sean M. Kinahan, David B. Silcott, Blake E. Silcott, Ryan M. Silcott, Danielle N. Rivera, Kevin K. Crown, Gabriel A. Lucero, Joshua L. Santarpia.

**Project administration:** Sean M. Kinahan, David B. Silcott.

**Resources:** Steven L. Distelhorst, Joshua L. Santarpia.

**Supervision:** Joshua L. Santarpia.

**Validation:** David B. Silcott, Blake E. Silcott, Ryan M. Silcott, Peter J. Silcott, Braden J. Silcott.

**Visualization:** Sean M. Kinahan, David B. Silcott, Blake E. Silcott, Ryan M. Silcott, Peter J. Silcott, Braden J. Silcott, Joshua L. Santarpia.

**Writing – original draft:** Sean M. Kinahan, David B. Silcott.

**Writing – review & editing:** David B. Silcott, Joshua L. Santarpia.

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
