## [Decision Letter · Decision Letter 0]

19 Apr 2021

PONE-D-21-03983

Aerosol tracer testing in Boeing 767 and 777 aircraft to simulate exposure potential of infectious aerosol such as SARS-CoV-2

PLOS ONE

Dear Dr. Kinahan,

Thank you for submitting your manuscript to PLOS ONE. After careful consideration, we feel that it has merit but does not fully meet PLOS ONE’s publication criteria as it currently stands. Therefore, we invite you to submit a revised version of the manuscript that addresses the points raised during the review process.

Thank you for submission of this very important manuscript.

Four experienced aerosol scientists (2 from US and another 2 from outside US) have agreed to review your manuscript.

All reviewers feel very positive about your manuscript although some minor revisions are recommended.

Those recommendations include: discussion on virus inactivation rate, quality of some of the figures, discussion on actual particle size that could contain virus in real-world (theoretically droplet containing single virus may dry down to ~100 nm but till it reached ~50 nm Brownian diffusion still should be very moderate therefore I would consider 1 micron size that you used for testing as appropriate but it should be discussed), mask fitting (mannequin vs real humans face) and more details on masks used (type, manufacturer), status of personal air-vents (In supplemental materials gaspers ON or OFF Columns are presented in the Tables, but there is not enough info about gaspers status in the main text. So, I would agree with this question and would like to have it more clearly written in the main text), location of IBAC sampling inlets (envisioning this question I've tried to clarify it with you prior to send manuscript to reviewers, but it seems like some more clarification is still needed).

Important question has been asked if your results are applicable to other environments (such as restaurants) and although this is of course out of scope of your work but recommendations on conducting similar studies in other populated environments could be discussed.

There are other minor comments/questions that you can find while reading each reviewer feedback.

So, I would like to request these minor revisions and hope after these revisions been made we can proceed with publication.

We look forward to receiving your revised manuscript.

Kind regards,

Vladimir Mikheev

Academic Editor

PLOS ONE

Journal Requirements:

"This work was funded through Zeteo Tech, Inc. by United States Transportation Command (USTRANSCOM - https://www.ustranscom.mil/).

Defense Advanced Research Projects Agency (DARPA - https://www.darpa.mil/). reviewed test plans and USTRANSCOM reports for editing and technical direction"

We note that you received funding from a commercial source: Zeteo Tech, Inc.

We note that one or more of the authors are employed by a commercial company: S3I LLC.

3.1. Please provide an amended Funding Statement declaring this commercial affiliation, as well as a statement regarding the Role of Funders in your study. If the funding organization did not play a role in the study design, data collection and analysis, decision to publish, or preparation of the manuscript and only provided financial support in the form of authors' salaries and/or research materials, please review your statements relating to the author contributions, and ensure you have specifically and accurately indicated the role(s) that these authors had in your study. You can update author roles in the Author Contributions section of the online submission form.

3.2. Please also provide an updated Competing Interests Statement declaring this commercial affiliation along with any other relevant declarations relating to employment, consultancy, patents, products in development, or marketed products, etc.  

5. Thank you for submitting the above manuscript to PLOS ONE. During our internal evaluation of the manuscript, we found significant text overlap between your submission and the following previously published works.

- https://images.radio.com/connectingvets/TRANSCOM%20Report%20Final.pdf

We would like to make you aware that copying extracts from previous publications, especially outside the methods section, word-for-word is unacceptable, even for works which you authored. In addition, the reproduction of text from published reports has implications for the copyright that may apply to the publications.

Please revise the manuscript to rephrase the duplicated text, cite your sources, and provide details as to how the current manuscript advances on previous work. Please note that further consideration is dependent on the submission of a manuscript that addresses these concerns about the overlap in text with published work.

Reviewers' comments:

Reviewer's Responses to Questions

**Comments to the Author**

1. Is the manuscript technically sound, and do the data support the conclusions?

Reviewer #1: Yes

Reviewer #2: Yes

Reviewer #3: Yes

Reviewer #4: Yes

2. Has the statistical analysis been performed appropriately and rigorously? 

Reviewer #1: Yes

Reviewer #2: I Don't Know

Reviewer #3: Yes

Reviewer #4: Yes

3. Have the authors made all data underlying the findings in their manuscript fully available?

Reviewer #1: Yes

Reviewer #2: Yes

Reviewer #3: Yes

Reviewer #4: Yes

4. Is the manuscript presented in an intelligible fashion and written in standard English?

Reviewer #1: Yes

Reviewer #2: Yes

Reviewer #3: Yes

Reviewer #4: Yes

5. Review Comments to the Author

Reviewer #1: Dear authors,

Major remark.

You used different tracers to study aerosol propagation in real aircrafts environments. But viruses in aerosols lose their biological activity at a rate that is a function of temperature, relative humidity, and other parameters. Therefore, the measurements carried out by the authors give only an upper estimate of the biological activity of the virus containing aerosol at the measurement points. This remark does not affect the conclusions drawn in the article about the low probability of passengers’ infection even in the nearest seats, however, I believe that the text of the article should contain a paragraph or two on the inactivation of SARS CoV-2 in aerosols, especially since information on the dependence of the inactivation rate of this virus on the above parameters are already presented in the literature.

Minor remarks:

L48: there should be reference (2) instead of (1);

L563: please remove the journal's name after paper authors

Figs 2 and 3 are presented in the paper in very bad quality.

Reviewer #2: This is a well-designed, elaborate study to measure the exposures associated with aerosol transmission from a single source within a Boeing 767 and 777 aircraft. While the results may be somewhat predictable from modeling airflow in these environments these are dramatic experimental data showing generally less than 0.1% at seats that are very close in proximity to the source of aerosol transmission. My specific comments are directed mostly at clarity and some further discussion of limitations in terms of particle size and use of face masks.

1. Abstract. “We completed over 65 releases of 180,000,000 fluorescent particles …” This seems unnecessary and likely not that precise, e.g. 65 releases of 180M each or all together? At least change to exponential notation.

2. Page 7. “2.2. 777-200 Testing. During the two days of in-flight testing, fluorescent tracer particles were released in the AFT, MID AFT, FWD, and FWD-MID sections of the airframe for a total of 64 releases. The breathing releases included 40 tests with the mannequin not wearing a mask and 24 tests with a mask.” This seems critical to understanding your methods/design and thus may need some clarification at this point. Fluorescent tracer particles nebulized? With what nebulizer? Were they fluorescent beads or solution? You give more specifics on these later in methods but some of the basic information would be helpful here.

3. Page 8. “The test sizes were chosen…” The sizes you chose to test are supported by your references and appropriate for your sampling methodology, APS, for particles larger than 0.5um but some data also show that the size of the Covid-19 virus ranges from 75 to 200 nm (e.g. Chen N et al. Lancet 2020;395:507), outside the range of the APS (i.e. may need SMPS and/or CPC to count these smaller particles). It’s clear that you can’t test the full range of particle sizes that may contain virus but you should include the potential for smaller sizes in your discussion of limitations associated with your choices. The smallest airborne viruses would likely have different mobility than 1um particles, i.e. the former more diffusive.

4. Page 11. “Standard surgical 3-ply masks …” What was type/manufacturer of mask? These may vary in terms of filtration efficiencies across manufacturers and fit to the face (especially a mannequin).

5. Page 16 and Table 2. 767 has better filtration than 777 despite having lower ACH? Other differences to explain this?

6. Page 18 and table 3. “As further evidence of the mannequin’s mask redirecting particles rather than filtering them …”. Again this may be due to poor mask fitting for the specific mask used and the fact that it was on a mannequin as opposed to an actual face. Should state as limitation of this aspect of the study.

Reviewer #3: This is a well-conducted and complicated study analyzing the horizontal spatial dispersion of particles meant to simulate those produced during normal breathing on two common intermediate-to long-trip commercial aircraft. The results show relatively low dispersion in both amount and distance, and validates the ventilation plans in these aircraft. Although this study is in response to the pandemic of the Covid 19 virus, it is generic enough to be relevant to nearly all respiratorilly generated communicative diseases. However, it does not take into account coughing and sneezing, which may or may not be the predominant form of transmission.

Review comments

1. Page 2, line 30, Abstract. The 65 releases are hard to associate with the reported 85 releases for the 767 and 64 releases of the 777 listed on page 7, Methods.

2. Page 5, line 107. This sentence announces over 300 releases.

3. Page 6, line 125. Were personal air vents on or off? My experience indicates most passengers usually have them on for varying flows. This can likely disturb flow from seat to seat or reduce particle count within the cone of air flow.

4. Page 7, line 59. 767-300 testing. Occasional power loss. When? How many times? During testing? Were tests repeated? Are these included in the 85 releases?

5. Page 7, why are there 64 releases for 777 and 85 releases for 767? Specific test configurations/seating are difficult to determine: i.e., were there more than triplicate tests per seat or other tests? For instance, for the 767, there were 9 seats listed for triplicate, both mask and no mask tests. That adds up to 9 (seats)X3 (triplicate)X2 (masks)=54 tests. Yet 85 releases reported.

6. Page 9, line 207. I am unfamiliar with IBACS. Were the IBAC sensor inlets located near typical face level and position?

7. Figure S2. I think I can only count 2 IBACS in the image although 4 is mentioned.

8. Could the volume taken up by actual persons in the seats change the results significantly? For instance, if a person takes up 25% of the volume in the sampled area, the counts would be the same, but concentration necessarily increases proportionally. And it may be important that concentration, in addition to counts, is important to infection.

9. Page 20, line 448-454 Discussion should mention status of personal air vents.

Reviewer #4: Review

Ms. Ref. No.: PONE-D-21-03983

Title: Aerosol tracer testing in Boeing 767 and 777 aircraft to simulate exposure potential of infectious aerosol such as SARS-CoV-2

Plos one

General comments:

This manuscript presents a study of aerosol release inside two aircrafts simulating potential exposure to SARS-CoV-2 viruses. The study is very topical and important. I recommend publication after my comments have been taken into account. These general comments are mainly my questions and opinions after reading the manuscript and any additions in these directions could help making the study more impactful. However, I let the authors decide how they can best respond to these comments.

Overall, this study is interesting, it is well designed and conducted, and provides important results concerning potential exposure to any air pollutants within an aircraft. As a first comment, I would like to see a critical disclaimer concerning the validity of this study in predicting the exposure to SARS-CoV-2 viruses. It is known from the literature that the size of SARS-CoV-2 virus is of the order of 120 nm. However, in some situations these viruses may exist as larger particles/droplets containing many viruses together with other material from sneezing, coughing, etc. Since the experiments of this study have been conducted using 1 and 3 �m particles, they do not exactly simulate the virus particles. In addition, the particle behaviour in an airflow and entering the sampler inlet is size dependent and behaviour of the two extremes – 120 nm and 3 �m – is very different and determined by different processes such as gravitational settling or inertia. These discussions could be added in the Discussion section to avoid too straightforward conclusions from these results to real life exposure.

It may be out of the scope of this study, but I remained curious to know whether the main results are something untypical in exposure scenarios or are these results similar to other potential exposure situations. In other words, it is expected that any point release of particles will be diluted as it enters the space in question and the question is if the reductions measured in this study indicate that the transport of viruses and consequently the exposure is insignificant. This study shows a minimum reduction of 99.54% of 1 μm aerosols from the index source to the breathing zone of a typical passenger seated directly next to the source. Would this value be similar or different in e.g. a restaurant situation with two people sitting next to each other?

Detailed comments:

The units: This is probably somewhat European perspective, but I find it always somewhat confusing that a mixed set of units are used. Naturally, it is the Journal policy, that determines the preferred unit system. I list here some of my observations:

-Line 126: The volume is given as cubic feet.

-Line 172: The particle concentration is given as “particles per liter of air”. Here the authors use different volume unit than previously. The abbreviation (pla) is unknown for me. This is used e.g. in line 200.

-Line 203: The instrument sampling rate is given as liters per minute (lpm).

-Line 236: The volume is given as milli liter, mL. Now the unit is written with capital L.

All of these are easy to understand, but perhaps some systematic way of using different units and symbols could be used.

Mask: I concluded that the mask was used for the mannequin that was the source of the particles. However, when reading the manuscript it was not immediately clear whether the source or the target was wearing the mask. This confusion is probably due to the fact that the discussion concerning the mannequin and the mask started already line 150, before the particle release process had been introduced (it comes rather late at chapter 2.8.).

The release process and the role of mannequin, chapter 2.4: The role of mannequin remained unclear. If I understand correctly, the aerosol release took place through the mannequin mouth/nose. However, the details remained unclear.

The size distributions, line 175 and supplemental fig 3: It is unclear what is presented in this figure. The y-axes unit for a size distribution is the size distribution function, not the concentration. The size distribution function is the concentration in given size bin normalized by the width of the bin, i.e. #/(cm3*(Dpmax-Dpmin)) which gives the unit for y-axes #/(cm3*�m). If the authors use the commercial instrument software this unit is probably the correct one automatically provided.

What is “ta” in line 179?

The sentence in lines 185-187 if difficult to understand.

The mask tests, chapter 3.2: What is the particle size used in these tests? If both particle sizes were used, was there any difference in the resulting reduction?

6. PLOS authors have the option to publish the peer review history of their article (what does this mean?). If published, this will include your full peer review and any attached files.

Reviewer #1: No

Reviewer #2: No

Reviewer #3: **Yes: **Kirby L Zeman

Reviewer #4: **Yes: **Kaarle Hämeri

---

## [Author Response · Author response to Decision Letter 0]

12 Jul 2021

Please see attached response letter for a point by point acknowledgment. Essentially all recommended changes were incorporated.

---

## [Editor Report · Decision Letter 1]

20 Sep 2021

PONE-D-21-03983R1

Aerosol tracer testing in Boeing 767 and 777 aircraft to simulate exposure potential of infectious aerosol such as SARS-CoV-2

PLOS ONE

Dear Dr. Kinahan,

Thank you for submitting your manuscript to PLOS ONE. After careful consideration, we feel that it has merit but does not fully meet PLOS ONE’s publication criteria as it currently stands. Therefore, we invite you to submit a revised version of the manuscript that addresses the points raised during the review process.

Please see my detailed comments below.

We look forward to receiving your revised manuscript.

Kind regards,

Vladimir Mikheev

Academic Editor

PLOS ONE

Journal Requirements:

Additional Editor Comments (if provided):

Dear Dr. Kinahan,

Thank you for revising the manuscript and addressing reviewer’s comments.

I think your manuscript is now in a much better shape but several minor edits/corrections are still required, please see below.

Line numbers were used from the “Revised Manuscript with Track Changes” version:

L52: “While large-droplet transmission likely does not vary substantially from the unique circumstance of being on an aircraft, the aerosol exposure route is dependent on the ventilation, environment”

Indicate please size of “large droplets”.

L160: tests with a mask. Methods sections 2.4, 2.5, and 2.8 describe aqueous tracer solutions and

You don’t have these 2.4, 2.5, and 2.8 numbering anymore.

L180: “minutes, and then placed into a static mode without airflow. The test solution, which is a 10% dilution of the stock solution in deionized water, was nebulized identically to in-flight tests, from a nebulizer plumbed to a mannequin (fluorescent tracer) with or without a mask, or without a mannequin (DNA-tagged tracer) and then briefly mixed using two remote-controllable fans (20-25 seconds).”

What was the stock solution?

Fluorescent tagged microspheres were aerosolized for one minute in a breathing pattern for 2 seconds on and 2 seconds off, correct?

L190: “for each release condition has low standard error around the mean, with a total number of fluorescent tracers when unmasked of 1.8*10^8, compared to 2.4*10^7 beads for the larger 3 µm DNA-tagged tracers (Table 1)”.

In the Table 1 you have both with and without mask results for fluorescent tracers but not in the text. I would suggest to add results for with masks (in the brackets): “for each release condition has low standard error around the mean, with a total number of fluorescent tracers when unmasked of 1.8*10^8 (with mask 1.7*10^8) compared to 2.4*10^7 beads for the larger 3 µm DNA-tagged tracers (Table 1)”.

L206: “sizes become increasingly sub-micron the effects of gravity vs Brownian motion”

For submicron particles its not a gravitational force – its inertial force. So I would change it to:

“sizes become increasingly sub-micron the effects of inertial forces vs Brownian motion”

L229: “IBAC sensors were placed in individual seats, with either 1 foot tubing extensions to actively sample within the breathing zone or placing on a collapsible crate to achieve the same sampling location”

How long was the sampling time?

L235: “for isokinetic sizing and alignment with a flow field”

Isokinetic sampling (not sizing)

L240: “typical resting passenger minute volume (7.5 lpm) rather than the instruments sampling rate of”

I think you need to clarify respiratory minute volume

L263: “(Sartorius), which operated for fifteen minutes, and collects 99.9995% of particles”

Need to specify instrument specs: what particle size range it collects

L265: “Five high volume air collectors were distributed near release”

Are those the same Airport MD8 or different air-samplers?

L289: “minutes and fluorescent tagged microspheres were generated aerosolized for one minute in a breathing pattern for 2 seconds on and 2 seconds off”

That is for both in the Chamber and on the plane tests, correct?

L316: “Economy (with and without a mask) and first class penetration percentages for nearby seats on the”

Although you previously explained very clear that terms “with and without a mask” relate to mannequin release of particles, some readers might still be confused and may think that here you are talking about mask protection for inhalation. I understand that for attentive reader it should be clear that you didn’t put mask on IBAC, but some people may still think that you somehow assessed mask inhalation protection. To avoid this potential confusion, I would suggest something like: “Economy (mannequin particles release with and without a mask)”. And for the Table 2 instead of “Breathing w/o mask or w/mask” I would also recommend to use “mannequin particles release with and without a mask”. Same comment for Fig 2 Caption: term “unmasked breathing” might be confusing.

Supplemental Tables 7 and 8 include “coughing” test conditions – What does it mean? I don’t think there is any mentioning about testing “coughing conditions” in the text.

Once again: What I didn’t find is how long was the IBAC sampling after each release. 15 min (same as Airport MD8 aerosol samplers)?

I also think that short discussion about how your data correspond to the real-world conditions is required. Please see below some thoughts:

According to Asadi 20201: “certain individuals are “speech superemitters” who emit an order of magnitude more aerosol particles than average, about 10 particles/second. A ten-minute conversation with an infected, asymptomatic superemitter talking in a normal volume thus would yield an invisible “cloud” of approximately 6,000 aerosol particles that could potentially be inhaled by the susceptible conversational partner or others in close proximity”. Assuming this is an approximation of upper limit we may estimate that during one minute talk this supermitter will release ~1,000 particles. It means that according to your measurements ~0.5% could reach the breathing zone of a typical passenger seated directly next to the source, or ~5 particles (for one minute release). For all other passengers it will be 0.02 - 0.03 % or 0.2 – 0.3 particles. Although infectious dose for COVID-19 is not known (at least I am not aware if any published literature data have this info), it might be as low as one particle. Therefore, sitting next to superemitter might not be safe.

1. Asadi, S., et al., The coronavirus pandemic and aerosols: Does COVID-19 transmit via expiratory particles? 2020, Taylor & Francis.
---

## [Author Response · Author response to Decision Letter 1]

7 Oct 2021

Editorial Board

PLOS ONE

October 1, 2021

Editorial Board: 

I am writing this letter to address the questions and revisions raised by the academic editor and reviewers for our most recent version of the manuscript entitled, “Aerosol tracer testing in Boeing 767 and 777 aircraft to simulate exposure potential of infectious aerosol such as SARS-CoV-2.” 

A point-by-point rebuttal appears below, as an appendix, as we did during the last revision. Overall, we attempted to accept, revise, and clarify at each point brought up in the review, and have responded below in blue ink. These changes were all relatively minor but important. We have also updated references that were preprints to their peer-reviewed final versions and fixed some non-Vancouver formatting.

Sincerely,

Sean Kinahan

National Strategic Research Institute

University of Nebraska Medical Center

42nd and Emile St. Omaha, NE 68198

skinahan@nsri.nebraskaresearch.gov

Tel: 1-443-255-3188

---

## [Editor Report · Decision Letter 2]

11 Oct 2021

Aerosol tracer testing in Boeing 767 and 777 aircraft to simulate exposure potential of infectious aerosol such as SARS-CoV-2

PONE-D-21-03983R2

Dear Dr. Kinahan,

We’re pleased to inform you that your manuscript has been judged scientifically suitable for publication and will be formally accepted for publication once it meets all outstanding technical requirements.

Kind regards,

Vladimir Mikheev

Academic Editor

PLOS ONE

Additional Editor Comments (optional):

Dear Dr. Kinahan,

Thank you for submitting the revised version of your manuscript.

I think this manuscript now can be accepted.

I have just one editorial note, please see below.

Thank you,

Vladimir

L122: “to more closely mimic coughing (Supplement)”.

Need to indicate Supplemental Tables Numbers S7 and S8

Also, throughout the text you are using term “Supplemental”, whereas Supplemental File is named as “Supporting Information”. Please be consistent with naming.
---

## [Editor Report · Acceptance letter]

5 Nov 2021

PONE-D-21-03983R2 

Aerosol tracer testing in Boeing 767 and 777 aircraft to simulate exposure potential of infectious aerosol such as SARS-CoV-2 

Dear Dr. Kinahan:

I'm pleased to inform you that your manuscript has been deemed suitable for publication in PLOS ONE. Congratulations! Your manuscript is now with our production department. 

Kind regards, 

on behalf of

Dr. Vladimir Mikheev 

Academic Editor

PLOS ONE